# The Evaluation of Bacterial Abundance and Functional Potentials in the Three Major Watersheds, Located in the Hot Spring Zone of the Tatun Volcano Group Basin, Taiwan

**DOI:** 10.3390/microorganisms10030500

**Published:** 2022-02-23

**Authors:** Viji Nagarajan, Hsin-Chi Tsai, Jung-Sheng Chen, Bashir Hussain, Cheng-Wei Fan, Aslia Asif, Bing-Mu Hsu

**Affiliations:** 1Department of Earth and Environmental Sciences, National Chung Cheng University, Chiayi 621, Taiwan; mathumitha08@gmail.com (V.N.); bashir.aku@gmail.com (B.H.); cwfan@eq.ccu.edu.tw (C.-W.F.); jabeen_dainic01@yahoo.com (A.A.); 2Department of Psychiatry, School of Medicine, Tzu Chi University, Hualien 970, Taiwan; cssbmw45@gmail.com; 3Department of Psychiatry, Tzu-Chi General Hospital, Hualien 970, Taiwan; 4Department of Medical Research, E-Da Hospital, Kaohsiung 824, Taiwan; nicky071214@gmail.com; 5Department of Biomedical Sciences, National Chung Cheng University, Chiayi 621, Taiwan; 6Doctoral Program in Science, Technology, Environment and Mathematics (STEM), National Chung Cheng University, Chiayi 621, Taiwan

**Keywords:** bacterial abundance, functional prediction, hot spring, Tatun volcanic group, waterbodies, 16S rRNA gene sequencing

## Abstract

The Tatun Volcanic Group (TVG), located in northern Taiwan, is characterized by acidic hot springs where the outflow of the hot springs may affect the properties of the associated lotic water bodies. We investigated the bacterial diversity and functional profiles of the Peihuang (PHC), HuangGang (HGC), and Nanhuang Creeks (NHC) located in the TVG basin using 16S rRNA gene sequencing coupled with statistical analyses. Water samples were collected from various streams of the creeks for two months of the year. The NHC showed the highest diversity, richness, and a unique number of phyla, which was followed by the HGC. A reduced number of phyla and a lower diversity was noticed in the PHC. The NHC was found to be abundant in the genera *Armatimonas*, *Prosthecobacter*, *Pirellula*, and *Bdellovibrio*, whereas the HGC was rich in *Thiomonas*, *Acidiphilium*, *Prevotella*, *Acidocella*, *Acidithiobacillus*, and *Metallibacterium.* The PHC was abundant in *Thiomonsa, Legionella*, *Acidocella*, and *Sulfuriferula*. The samples did not show any strong seasonal variations with the bacterial diversity and abundance; however, the relative abundance of each sampling site varied within the sampling months. The iron transport protein- and the sulfur metabolism-related pathways were predicted to be the key functions in all the creeks, whereas the heavy metal-related functions, such as the cobalt/nickel transport protein and the cobalt–zinc–cadmium efflux system were found to be abundant in the HGC and PHC, respectively. The abundance of *Bdellovibrio* in the NHC, *Diplorickettsia* in the HGC, and *Legionella* in the PHC samples indicated a higher anthropogenic impact over the creek water quality. This study provides the data to understand the distinct bacterial community structure, as well as the functional potentials of the three major watersheds, and helps the knowledge of the impact of the physicochemical properties of the TVG hot springs upon the watersheds.

## 1. Introduction

Hot springs are considered extreme environments because of their unique geochemical characteristics [1,2]. They are mostly associated with high volcanic activity sites and are heated by geothermal energy [3]. Hot springs are termed the “tropical rain forests” of the microbial world [4]. Hot springs act as a natural laboratory to explore a wide range of microbial communities and to study their adaptation strategies to survive extreme conditions [5]. Microorganisms play a key role in the biogeochemical cycling of major elements [6] in hot spring-associated ecosystems; thus, it is important to examine their microbial community structure. The microbial diversity in hot springs is dictated by the environmental parameters and physicochemical characteristics of the water [7]. Metagenomic analyses have described microbial communities and their functionality in extreme environments [8]. Hot springs in river basins may be channeled directly into proximal lotic water bodies, such as rivers and creeks. This results in hot water mixing with the river water, altering their physicochemical characteristics and microbiota [9]. The water bodies harbor hot spring-native microbial populations, along with introduced potential pathogens [10,11], due to anthropogenic influences.

Our study sites included three major watersheds, the Peihunag (PHC), Huanggang (HGC), and Nanhuang creeks (NHC), originating from the Tatun Volcano Group (TVG) in north Taiwan. The TVG has extensive post-volcanic activities and comprises a series of andesitic volcanoes. Many hot springs and fumaroles have been found in this volcano group and are primarily of meteoric origin [12]. The two major faults in the TVG are the Kanchiao and Chinshan faults, and the hot springs are mainly distributed along the Chinshan fault [13]. The major hot springs located in the TVG are as follows: Ti-re-ku, Liou-huang-ku, Tingbiqiao, Hoshan, Long-feng-ku, Matsao, Lengshniken, Qigu, Tayukeng, Bayan, Szehuangping, and Tapu [13,14]. Among the aforementioned hot springs, the Long-feng-ku (PHC), Ti-re-ku, Liou-huang-ku (HGC), and Bayan (NHC) hot springs are confluent with our three study watersheds [12,15]. The hydrogeochemical characteristics of the hot spring water associated with TVG are well understood [13,14]. The hot springs of the TVG can be divided into three categories according to their hydrogeochemical characteristics: acidic sulfate, chloride-rich acidic sulfate, and weakly acidic to neutral bicarbonate hot springs [16]. Previous studies have shown that the hot spring waters of the TVG are heavily contaminated with arsenic, sulfur, iron, and manganese [14,15], which may pollute water systems, such as rivers and creeks originating from nearby foothills. Thus, understanding the characteristics of the TVG-associated hot springs may help to determine the microbial features of nearby water bodies. Indeed, the associated water bodies may be inhospitable for common microbiota because of the influence of the hot springs. Therefore, it could be of great interest to explore the distinct microbial diversity of the water bodies, which are influenced by the TVG hot springs.

The first study site, the PHC (north sulfur creek), is the main watershed system in the TVG [17], named for its location and yellow color caused by high sulfur content. The two major tributaries, the Lujiaoken Creek (western side) and the Matsao Creek (southern side), converge to form the main stream of the PHC in the Bayan zone, where the creek enters the plains area [18]. Despite the major hot springs in this zone, their discharge is far from the creek channel [18]. Nevertheless, because of the mixing of hot spring water with the creek, the water is characterized as acid sulfate thermal water (pH 3–4.5; SO_4_ 60–400 ppm) with a high concentration of iron, aluminum, and silica in the form of iron-aluminum hydroxide [17]. The high concentration of metal elements in the creek water favors the formation of suspended colloids, which absorb the dangerous levels of heavy metals, such as chromium, arsenic, and lead [17]. The second study site, the HGC, is one of the major watersheds in the Guandu Plain in northern Taiwan [12]. It originates from the sulfur valley (Liou-huang-ku) and flows southward through the Beitou thermal valley. One of the tributaries of the HGC originates from the geothermal valley (Ti-re-ku), which converges with the main stream and flows along the Keelung River [12], which may impact the water chemistry of the HGC. The creek water was characterized as acid sulfuric (pH < 4.0) with a high concentration of various metals, such as iron, sulfide, aluminum, manganese, arsenic, and lead [19]. The presence of trace elements and sulfate contaminants in the creek may be caused by the outflow of the associated hot springs [12]. The third study site, the NHC, originates from Bamboo Lake between Xiaoguanyin Mountain and Seven-Star Mountain (Chi-Shin Mountain) and flows through the Guandu Plain. The creek is confluent with hot springs in the dragon phoenix valley (Long-feng-ku) [20]. The dragon phoenix valley has been reported to be slightly acidic (pH 5.7) and rich in sulfur content [13].

Although the hydrogeochemical characteristics of the TVG hot springs are well documented, to the best of the authors’ knowledge, no report describes their influence on the microbial community structure of the proximal water bodies. We hypothesize that the outflow of the TVG-associated hot springs in the major watersheds may shape the microbial community structure and metabolic functions, which might cause complex changes in water chemistry. The lotic water bodies examined in this study are predominantly used for localized agricultural irrigation and other allied activities; thus, we attempted to explore the microbial abundance and functional potential of these creeks. In this context, the goal of this study was to uncover and compare the bacterial diversity and functional metabolic profiles of water samples from the three major watersheds, including the HGC, PHC, and NHC, located in the TVG site. We performed amplicon sequencing of the 16S rRNA genes, using high-throughput sequencing to analyze the bacterial community structure. Thus, this study extensively elucidates the microbial community structure and metabolic potential of major watersheds located in the hot spring zones of the TVG basin.

## 2. Materials and Methods

### 2.1. Sample Site Description and Sampling Details

The three studied watersheds, namely, the PHC, HGC, and NHC, originate from the TVG in northern Taiwan. The geographical location of the three watersheds is shown in Figure 1. A total of 17 water samples, including 6 from the PHC, 6 from the HGC, and 5 from the NHC have been collected accordingly. The sampling was done in March and July 2019, and the sampling details are given in Appendix A. One liter of water samples were collected from each of the sampling sites using a sterile sampling cup and were poured into a sterile zip-lock sampling bag. The temperature of the samples was measured by a portable thermometer in real-time (HI9828, Hanna Instruments, Woonsocket, RI, USA). Furthermore, the collected samples were labelled accordingly and were transported to the laboratory under a controlled temperature for subsequent molecular analyses.

### 2.2. Physicochemical Properties of the Sampling Sites

The physicochemical properties of the water samples of the watersheds, namely, the PHC, HGC, and NHC, as well the TVG-associated hot springs, were collected from the past investigations. The details are shown in Appendix A. 

### 2.3. Microbial DNA Extraction

The water samples were filtered through a GN-6 filter membrane (GN-6 Metricel^®^, PALL Corporation, New York, NY, USA) with a pore size of 0.4 µm. Fifty milliliters of a phosphate-buffered saline (PBS) solution was used to elute the filter material on the filter membrane. The elute was then centrifuged at 2600× *g* for 30 min in a high-speed centrifuge and the supernatant was discarded. Four hundred microliters of a lysis buffer (ZP02006 Kit, lysis buffer, Taipei, Taiwan) was added with the obtained pellets to break the bacteria. The digested lysate was extracted with an automatic nucleic acid extraction instrument, the MagPurix^®^ Auto DNA Extraction Kit (ZP02006) (ZINEXTS Life Science Corp., New Taipei City, Taiwan) following the manufacturer’s protocol. Later, the quality of the microbial gDNA was assessed by running gel electrophoresis (1.5% gel in a Tris-acetate ethylenediaminetetraacetic acid buffer). The concentration of the extracted microbial gDNA was measured using the NanoDrop 2000 spectrophotometer (Thermo Fisher Scientific Inc., Wilmington, DE, USA). Then, the purified gDNA was stored at −20 °C for further analyses.

### 2.4. 16S rRNA Amplicon Sequencing, Library Construction, and Data Analyses

The hypervariable V3-V4 region of the 16S rRNA gene was amplified using the following set of forward and reverse primers: Pro341F (5′–CCTACGGGNBGCASCAG–3′) and Pro805R (5′-GACTACNVGGGTATCTAATCC-3′) [21]. The PCR was carried out in a thermocycler (Px2 Thermal Cycler, Thermo, Waltham, MA, USA) with a total volume of 25 μL that contained 3 μL of the template DNA, 2X KAPA HiFi HotStart Ready Mix (12.5 μL), 400 nM of each primer, and ddH_2_O (7.5 μL). The PCR reaction conditions were as follows: an initial denaturation at 95 °C for 30 s, followed by 28 cycles of denaturation at 95 °C for 30 s, with annealing at 55 °C for 30 s, and elongation at 72 °C for 30 s. The PCR products with the expected sizes were purified from the matrix after the separation using agarose gel electrophoresis. Then, the purified 16S rRNA gene amplicons were subjected to sequencing using the pair-end method with the MiSeq Illumina platform (Illumina Inc., San Diego, CA, USA), following the manufacturer’s instructions. The Nextera XT DNA Library Preparation Kit (Illumina) was used to construct libraries from the isolated DNA. The quantitative insights into microbial ecology (QIIME2) pipeline were then used to process the obtained raw Illumina MiSeq sequence data. Amplicon sequence variants (ASVs) were picked using DADA2 after filtering the chimeric sequences, marginal sequence errors, and noisy sequences. Furthermore, the taxonomy classification was done using the SILVA reference database [22]. The relative abundance of the microbial community associated with each of the samples was obtained using the QIIME2 view.

### 2.5. Metabolic Functional Prediction of the Bacterial Communities

The potential metabolic functions of the bacterial communities related to the three watershed sampling sites were determined using the latest version of the PICRUSt2 pipeline, using the Kyoto Encyclopedia of Genes and Genomes (KEGG) database [23]. The shift in predicted functions among the samples of the three watersheds were statistically analyzed using STAMP software [24]. Additionally, a Spearman correlation was performed to evaluate the significant correlations between the bacterial diversity and their predicted metabolic functions using IBM SPSS Statistics 24 (IBM, Armonk and North Castle, NY, USA).

## 3. Results

### 3.1. Sequence Depth, ASV Analysis, and Bacterial Diversity

A total of 8,459,300 sequence reads were observed, targeting the V3-V4 regions of 16S rRNA in the three watersheds samples. After quality filtering, a total of 295,628 high-quality sequences were retained, associated with the three watersheds that were assigned to ASVs using the SILVA database. The rarefaction of these sequences was performed at the lowest depth (4500) to compare the bacterial diversity among the three watersheds (Appendix A). A Venn diagram was generated to demonstrate the proportion of shared and unique ASVs present among the three watersheds (Appendix A). Among the three experimental groups, a higher number of ASVs were noted in the NHC (*n* = 2498), followed by the HGC (*n* = 1454), while the lowest number of ASVs was recorded in the PHC (*n* = 746). Likewise, the highest unique ASV diversity was found in the NHC (*n =* 2447), followed by the HGC (*n =* 1336) and the PHC (*n =* 672). However, only three ASVs were found to be common in the three of the watersheds. Comparatively, the PHC:HGC (*n =* 69) showed a greater number of common and shared ASVs than the HGC:NHC (*n =* 46) and the PHC:NHC (*n =* 2).

The distribution of bacterial diversity and richness indices, such as Chao1, the Shannon index, and the Simpson index were visualized among the samples of each waterbody, as shown in Figure 2. The higher alpha-diversity indices, such as Chao1 (Figure 2A), the Shannon index (Figure 2B), and the Simpson index (Figure 2C) were noted in the NHC, followed by the HGC. The results, therefore, revealed that the PHC had the highest diversity richness compared to the others. Amidst this, the lowest level of diversity richness was noted in the samples of the PHC experimental group. All the included alpha diversity indices showed significance among the experimental groups, following Kruskal–Wallis (*p* < 0.05) group comparisons. Additionally, a principal coordinate analysis (PCA) of the beta diversity results, based on Bray–Curtis distances, showed a discrete clustering pattern and a compositional dissimilarity among the three experimental groups. It is evidenced that the samples from the NHC, HGC, and PHC had unique and different bacterial communities (Figure 2D). The beta diversity analysis also showed a significant clustering pattern following a PERMANOVA analysis (*p* < 0.05). 

### 3.2. Bacterial Abundance Based on 16SrRNA Gene Amplicons

#### 3.2.1. Phylum Level Bacterial Abundance

A total of 35 bacterial phyla were noticed in the three watersheds in the two sampling months, and are shown in Figure 3. Proteobacteria was the top predominant phyla noticed in the three watersheds of the two sampling months. Among the NHC sampling sites, W01 (Xiaoyoukeng Yongquan) have been found to be dominant with the phylum Crenarchaeota, Nanoarchaeota, and Firmicutes, regardless of the sampling months. The remaining sampling sites of the NHC (W02, W03, W04, and W05) were dominant in Bacteroidota, Verrucomicrobiota, and Planctomycetota. All the samples belonging to the HGC site were found to be abundant with the phyla Bacteroidota, Cyanobacteria, Acidobacteriota, and Verrucomicrobiota, irrespective of the sampling months. Regarding the PHC, the phylum Cyanobacteria was found to be abundant in all the sampling sites in both the sampling months. Nevertheless, the sample W02 of the PHC, was found to. Be dominant in Nitrospirota and Actinobacteria, and W04 was abundant in Crenarchaeota, Aquificota, and Firmicutes.

#### 3.2.2. Genus Level Bacterial Abundance

A Venn diagram was drawn to demonstrate the proportion of unique and common bacterial genera present among the three watersheds (Figure 4). Consistent with the AVS analysis, a higher number of bacterial genera were recorded in the NHC (*n* = 283), followed by the HGC (*n* = 251), whereas the lowest number of genera were recorded in the PHC (*n* = 126). In addition, the highest percentage of unique genera diversity was found in the NHC (32.6%), followed by the HGC (17.2%) and the PHC (4.9%). Comparatively, the PHC:HGC (20.7%) showed a greater proportion of common genera than the HGC:NHC (11.9%) and the PHC:NHC (4%). However, only 8.6%, i.e., 37 genera were found to be common in all three watersheds. Furthermore, a heat map analysis was built to visualize the abundance and dominant bacterial communities at the genus level (Appendix A). A higher number of the bacterial genus was found to be abundant in the NHC, followed by the HGC and the PHC. However, considering the sampling months, there is not much difference in the clustering pattern that was noticed in the three watersheds.

The top 20 most dominant bacterial genera noticed in the three watersheds in the two sampling months are shown in Figure 5. Overall, the highest genus abundance was exhibited in the samples of the NHC site, followed by the HGC. The lowest genus abundance was noted in the PHC site, particularly in PHC-W01. The NHC sampling site was found to be dominant with the following genera, namely, *Armatimonas*, *Prosthecobacter*, *Pirellula*, *Bdellovibrio,* and *SH3-11*. The genera *Thiomonas*, *Acidiphilium*, *Prevotella*, *Acidocella, Acidithiobacillus*, and *Metallibacterium* were found to be dominant in the HGC. The PHC site was found to be abundant in the genera *Thiomonsa*, *Legionella*, *Acidocella,* and *Sulfuriferula*. The genus *Sediminibacterium* and *Novosphingobium* have displayed a ubiquitous abundance in three of the watersheds; nevertheless, the relative percentage of abundance was varied accordingly. Considering the sampling period, all three watersheds have shown an almost similar genera abundance in both months; however, the relative percentage of the abundance of each sampling site varied within the sampling months.

#### 3.2.3. LEfSe Analysis of Differential Bacterial Abundance

A biomarker analysis using the linear discriminant analysis (LDA) with effect size measurements (LEfSe) was carried out to determine the enriched taxa and to evaluate the variation in the bacterial abundance between the experimental groups. LEfSe detected seven bacterial clades showing statistically significant differences between the samples of the HGC and NHC sites (Figure 6A,B). The Pedosphaerales at the order level and *SH3_11* at the genus level had a significantly higher abundance in the NHC site. The branch diagram of the LEfSe highlighted the higher abundance of Pedosphaerales and *SH3_11.* In the HGC samples, Acidobacteriales, Diplorickettsiales, and Bacteroidales at the order level, *Diplorickettsiaceae* at the family level, and *Acidiphilium* at the genus level had significantly higher abundances. Moreover, the LEfSe highlighted the most abundant bacterial clades. 

### 3.3. Metabolic Functional Prediction of the Bacterial Community

#### 3.3.1. PCA and Shift Analysis

The potential metabolic functions of the bacterial community belonging to the three watersheds were determined using the latest version of the PICRUSt2 pipeline by the KEGG database. A PCA was done to group the samples based on the predicted metabolic functions of the predominant bacterial communities (Figure 7A). PC1, PC2, and PC3 accounted for 77.1%, 10.2%, and 5.9% of the total variation, respectively. The results demonstrated the clear distribution pattern and the uniqueness of the samples concerning their predicted metabolic functions. Comparatively, the samples of the PHC and HGC sites had fewer variations, indicating less dissimilarity in their predicted metabolic functions than the NHC site.

Additionally, the Welch *t*-test, following FDR (*p* < 0.05), was used to analyze the shifts in the predicted functional potentials of the bacterial communities to know the statistical significance among the experimental groups (Figure 7B–D). The analysis revealed that 21 predicted functions were significantly enriched in the NHC and HGC samples (Figure 7B). Among them, nine of the functions, such as the bifunctional oligoribonuclease and PAP phosphatase (K06881), ferrous iron transport protein A (K04758), FMN reductase (K00299), manganese/iron transport system permease protein (K11606), manganese/iron transport system substrate-binding protein (K11604), manganese/iron transport system ATP-binding protein (K11607), manganese/iron transport system permease protein (K11605, K09819), and the manganese/iron transport system ATP-binding protein (K09820) were significantly abundant in the NHC samples, whereas the mean proportion was lower in the HGC. Nonetheless, the remaining 11 functions, such as the cobalt–zinc–cadmium efflux system protein (K16264), iron (III) transport system permease protein (K02011), 3′(2′), 5’-bisphosphate nucleotidase (K01082), high-affinity iron transporter (K07243), bifunctional enzyme CysN/CysC (K00955), cobalt/nickel transport system permease protein (K02008), sulfur-oxidizing protein SO_X_X (K17223), cobalt/nickel transport protein (K02009), sulfur-oxidizing protein SO_X_B, SO_X_A (K17224, K17222), and the sulfane dehydrogenase subunit SO_X_C (K17225) were significantly enriched in the HGC samples.

Likewise, 18 metabolic functions were significantly enriched in the PHC and NHC samples (Figure 7C). The 14 major functions, such as the bifunctional oligoribonuclease and PAP phosphatase NrnA (K06881), molybdopterin-containing oxidoreductase family membrane subunit (K00185), molybdopterin-containing oxidoreductase family iron-sulfur binding subunit (K00184), ferrous iron transport protein B (K04759, ferrous iron transport protein A (K04758), O-acetylhomoserine (thiol)-lyase (K01740), FMN reductase (K00299), manganese/zinc/iron transport system ATP-binding protein (K11710), manganese/iron transport system permease protein (K11606, K11605, K09819), manganese/iron transport system ATP-binding protein (K11607, K09820), and the manganese/iron transport system substrate-binding protein (K11604) were significantly abundant in the NHC samples, whereas the mean proportion was lower in the PHC samples. However, the PHC samples were found to be significantly enriched with four of the functions, such as the cobalt–zinc–cadmium efflux system protein (K16264), thiosulfate/3-mercaptopyruvate sulfurtransferase (K01011), thiosulfate reductase (K08352), and the high-affinity iron transporter (K07243). The one metabolic pathway related to the sulfite reductase (NADPH) flavoprotein alpha-component (K00380) was significantly enriched between the PHC and HGC sites (Figure 7D). The abundance was higher in the HGC samples, while the mean proportion was decreased in the PHC. 

#### 3.3.2. Correlation between the Significant Bacterial Genera and Predicted Metabolic Functions

A Spearman correlation analysis was done to find the correlations between the statistically enriched bacterial genus and their predicted metabolic functions associated with the three watersheds (Figure 8). The correlation was considered significant at *p* < 0.05. The genera *Metallibacterium*, *Acidiphilium*, and *Thiomonas* showed a significant positive correlation with K02008 and K02009 (cbiQ; CbiN: cobalt/nickel transport system permease protein). The genera *Sediminibacterium*, *Novosphingobium*, *Pirellula*, *Prosthecobacter*, *Armatimonas*, and *SH3-11* were significantly and positively correlated with K03325 (ACR3, arsB; arsenite transporter), K11604 (sitA), K11605 (sitC), K11606 (sitD), K11607 (sitB), K09819 (manganese/iron transport system), K17225 (SO_X_C; sulfane dehydrogenase subunit), K17222 (SO_X_A; sulfur-oxidizing protein SOxA), K04758 (feoA; ferrous iron transport protein A), and K04759 (feoB; ferrous iron transport protein B). Additionally, *Sulfuriferula*, *SH3-11, Prosthecobacter, Sediminibacterium*, and *Novosphingobium* had a significant positive correlation with K01011 (Thiosulfate sulfurtransferase). The genera *Acidiphilium*, *Armatimonas,* and *Prosthecobacter* exhibited a strong positive correlation with the K00955 (cysNC; bifunctional enzyme), but it was not significant. Indeed, the majority of the predicted functions were significantly and positively correlated with uncultured sequences; however, the role of specific bacterial communities over the potential metabolisms remains obscure.

## 4. Discussion

A comprehensive examination of the microbial community structure of the three major watersheds originating from the TVG was conducted using 16S rRNA gene amplicon sequencing. An alpha diversity analysis revealed that the metrics varied greatly among the three watershed samples in both sampling months. The higher value of alpha diversity noted in the NHC indicated that the NHC samples were more unique and diverse than other watersheds. Following the NHC, a higher alpha diversity was noted in the HGC. Consistent with alpha diversity metrics, more ASVs were noted in the NHC, followed by the HGC. The lowest ASVs and metrics were noted in the PHC samples, which illustrated its lower diversity and richness than the NHC and HGC. Previous hydrochemical studies stated that the NHC water samples were categorized as moderately to slightly acidic (pH 4.2–6.3) [20], whereas the PHC water was characterized as highly acidic to moderately acidic (pH 3–4.5) [17]. Regarding the HGC, the upstream samples were slightly acidic (pH 5.4–6.5), whereas the midstream and downstream samples were highly acidic (pH 2.4–3.3). The pH was considered an obvious factor influencing the microbial communities in hot spring environments [25]. A moderate-to-slightly acidic range could be the reason for the higher bacterial abundance and diversity in the PHC and HGC water samples. In contrast, the highly acidic nature of the PHC samples may, in turn, have resulted in reduced bacterial abundance compared to the other watersheds. An increased bacterial abundance with an increasing pH was observed in our study, which showed a positive correlation between the pH and bacterial abundance. Moreover, compared to Ti-re-ku and Bayan, Long-feng-ku was reported to have a lower heavy metal content, including nickel, arsenic, and lead [14,15]. The outflow of the heavy metal-rich Ti-re-ku and Bayan hot springs may contaminate nearby water bodies, such as the HGC and PHC, respectively. Heavy metal contamination may alter the composition, activity, and metabolism of microbial communities through functional disturbances, such as cell membrane destruction, protein denaturation, and enzyme activity inhibition [26]. This may have resulted in the lower bacterial diversity and abundance in the HGC and PHC than in the NHC samples.

Among the observed bacterial phyla, Proteobacteria exhibited overwhelming dominance in all three watersheds, regardless of the sampling month. As stated in many previous studies, Proteobacteria appears to be indigenous to hot spring sites and is dominant in hot springs in various countries [27,28,29]. The phylum Crenarchaeota, which was abundant in the NHC, has been reported as a thermophilic sulfur-dependent acidophile [30] that is abundant in the sulfur-rich hot springs of volcanically active regions [31,32]. The higher SO_4_^2−^ content of Long-feng-ku [14] may, in turn, support the growth and abundance of Crenarchaeota in the NHC. The second most abundant phylum, Nanoarchaeota, may be correlated with the hypersalinity of the NHC, which is widely distributed in various hypersaline terrestrial hot springs [33]. Additionally, the sulfur-rich downstream samples of the NHC [20] were abundant in Planctomycetota. A previous study also found isolated Planctomycetota sequences from sulfur-rich hot spring sites [34]. The phylum Planctomycetota can survive in sulfur-rich environments through sulfur reduction functional pathways [34]. The photosynthetic phylum, Cyanobacteria, is an important phylum of hot spring assemblages, was abundant in the HGC and PHC samples. Moreover, it is a dominant primary producer of hot spring microbial mats and has been previously isolated from biofilms of various sulfur-rich hot spring sites [35,36]. The second most abundant phylum, Verrucomicrobia, is a thermo-acidophile that has been isolated from various hot spring locations [37]. However, the HGC water may be influenced by the highly acidic geothermal valley (Ti-re-ku, pH 1.8) and the sulfur valley (Liou-huang-ku, pH 2.7–4.2) [14]. Verrucomicrobia can survive better because of their thermal-acidophilic nature [37]. Likewise, the thermophilic bacterial phyla, Bacteroidota and Acidobacteriota, rich in the HGC samples, have been previously observed in various hot spring sites [38]. Regarding the PHC, the downstream samples showed a higher proportion of Nitrospirota, Actinobacteria, and Aquificota. The downstream of samples of the PHC may converge with the sulfur-rich hot spring of the Bayan site (SO_4_^2−^ 88.44 ppm) [15]. However, the bacterial communities, such as Nitrospirota, can exist in that extremity due to its sulfate-reducing metabolism [37]. Likewise, Aquificota was abundant downstream and has been previously isolated from many terrestrial hot ecosystems [39]. Because of the hyperthermal adaptations and the metabolically versatile behaviors [40,41], the phylum, Aquificota, can survive better in the hot spring convergence zone of the PHC.

A higher number of unique genera was found in the NHC samples, followed by the HGC and PHC, which corroborates the phylum results. The genus *Armatimonas*, dominant in the NHC samples, was previously isolated from the iron-rich sediments of thermophilic environments, including hot springs and geothermal fields [42]. The higher iron content of the middle reaches of the NHC [20] may support the growth and survival of *Armatimonas*, which is mainly involved in redox metal cycling in the thermal environment [43]. Another genus, *Prosthecobacter*, also rich in the NHC samples, may adapt to hot spring-influenced water bodies through the production of the tubulin protein [44]. The tubulin protein further polymerizes and forms microtubules that enable the intracellular transport of microbial cells as an adaptation behavior. The genus, *Pirellula,* was previously isolated from the sulfide-saturated sediments of mesophilic springs [34] and was abundant only in the NHC. The higher sulfur concentration of Long-feng-ku [14] may result in an increased abundance of the genus, *Pirellula,* in the NHC, compared to other watersheds. The genus, *Bdellovibrio*, also abundant in the NHC, was previously isolated from the arsenic-rich hot springs of the Himalayas [45]. Furthermore, *Bdellovibrio* has also been found in freshwater bodies [46], polluted rivers [47], and the human gut [48]. Thus, *Bdellovibrio* may be introduced to the NHC owing to the anthropogenic impact on the water body. 

The HGC samples had abundant *Thiomonas*, a mild thermophile, previously observed as the dominant bacteria in acidic hot springs [49] and arsenic-rich mine drainages [50]. The rich hydrogen sulfide and arsenic content of the geothermal valley (8589 and 398 μmol/mol) [51] may contaminate the nearby waterbody, the HGC. *Thiomonas* is a mixotrophic sulfur and arsenic oxidizer [52], which further facilitates biological deodorization and detoxification mechanisms of hydrogen sulfide and arsenic compounds, respectively [52,53]. Moreover, high acidity (pH 1.8), iron (86.3 ppm), and sulfate (2390 ppm) levels in the geothermal valley [14] result in a higher abundance of thermo-acidophilic *Acidiphilium* in the HGC. *Acidiphilium* is dominant in acidic hot springs [54] and is involved in sulfur- and iron-related metabolic functions [55]. *Prevotella* was higher in the HGC, which is a human gut bacterium, and has been previously isolated from human feces [56]. The HGC is a popular spa area in Taiwan [57]; thus, anthropogenic interventions in the creek may be the key reason for the abundance of *Prevotella*. In addition, *Prevotella* was less tolerant to dissolved oxygen content [58]. The reduced level of dissolved oxygen in the HGC (<5.0 mg/L) [20] may support the existence of *Prevotella* in the HGC. The acidophile, *Acidocella,* was observed in the HGC and has been reported to be tolerant to a high concentration of various metals, including aluminum [59]. The geothermal valley has been shown to have a high aluminum content [14], which leads to the aluminum contamination of the HGC. However, the genus *Acidocella* can tolerate aluminum toxicity, and the elevated aluminum content may protect *Acidocella* from the growth-inhibiting effect of an acidic environment [59]. The genus, *Acidithiobacillus*, which is abundant in the HGC, is an iron- and sulfur-oxidizing bacteria found in acidic environments of various geographical locations [60]. *Metallibacterium*, which is rich in the HGC, is an acid-tolerant anaerobic bacterium previously isolated from the acidic biofilms of pyrite mines [61]. 

Like the HGC, the higher sulfur and arsenic content of the Bayan zone (88.44 ppm and 0.30 ppb, respectively) [15] may contaminate the downstream reach of the PHC. This facilitates the abundance of *Thiomonas*, which is often involved in the sulfur and arsenic metabolism in hot spring sites [52]. Likewise, the higher aluminum content of the Bayan zone of the PHC (4.34 ppm) may contribute to the growth and survival of the aluminum-tolerant bacterial genus, *Acidocella,* in the PHC. In addition, *Legionella* was abundant in the PHC, and its abundance was correlated with the prevalence of other indicator microorganisms. Hot spring resorts offer bathing and spa services, which are popular vacation activities in Taiwan [62]. Hence, human activities may have introduced *Legionella* to the PHC, which has also been isolated from other hot spring resorts in Taiwan [63]. Additionally, the PHC-abundant *Sulfuriferula* was previously isolated from hot spring water in Japan [64]. The genus is a sulfur-oxidizing bacterium that may be responsible for the sulfur metabolism [64] at associated sites. The genus abundance results clearly demonstrated that the out-flow of hot springs highly influenced the bacterial abundance of the three associated watersheds, which may be caused by the interference of their physicochemical properties. Although the bacterial abundance varied among the watersheds according to the characteristics of the nearby hot springs, there was not much variation between the sampling months. 

The LEfSe analysis highlighted the abundance of Pedosphaerales (order) and SH3_11 (genus) in the NHC samples. Pedosphaerales has been isolated from freshwater bodies, such as lakes, in previous studies [65,66]. Moreover, Pedosphaerales has been observed to be the predominant rhizosphere microbiota of many crops [67,68], which indicates the existence of aquatic plant communities in the watershed. Thus, it is evident that the NHC can support the growth and survival of freshwater-related bacteria, such as Pedosphaerales, unlike other watersheds in our study. *Diplorickettsia* was detected in the HGC samples and is a human pathogen that causes tick-borne infections in humans [69,70]. The existence of human pathogens demonstrated a greater influence of human activities in the HGC than in the other two watersheds. In addition, *Acidiphilium* was abundant in the HGC, which is one of the iron-reducing heterotrophs observed in acidic hot springs [55]. Comparatively, the geothermal valley (pH 1.58) of the HGC was more acidic than the Long-feng-ku (pH 5.71) of the NHC [13]. Thus, the high acidity of HGC may facilitate the greater occurrence of *Acidiphilium* compared to the other watersheds. In addition, a geothermal valley was recorded with a higher iron concentration than the hot springs of the remaining watersheds [14]. The higher iron content may be sedimented in the watersheds and may serve as electron acceptors for iron-reducing bacteria, such as *Acidiphilium* [71]. Furthermore, the genera found to be common in the three watersheds, namely, *Sediminbacterium* and *Novosphingobium*, were isolated from hot spring sites in various geographical locations [72,73,74,75]. 

Considering the metabolic functions, iron and manganese transportation and iron–sulfur binding were predicted to be key functions in the NHC samples. As found in previous investigations, Long-feng-ku contains a higher concentration of elements, such as iron and manganese [14], which are highly correlated with the functional potential of the NHC samples. Under oxygen deprivation, bacterial communities in water bodies may develop evolutionary mechanisms for their survival. In particular, the genus *Prosthecobacter*, under anoxic conditions, can reduce the insoluble form of manganese to a soluble form and can use it as an alternative to oxygen [76,77]. This process leads to the further formation and addition of manganese oxides in the water bodies [78]. Similarly, chemolithoautotrophic bacterial communities may deprive iron oxidation for growth in iron-rich sites [77]. Thus, in the iron- and manganese-rich environment of the NHC, bacterial communities, such as those represented by *Prosthecobacter*, exhibited a higher abundance of iron- and manganese-related metabolism. The higher abundance of thermoacidophilic archaea communities, such as Crenarchaeota and Nanoarchaeota in the NHC, are involved in iron–sulfur binding-related metabolism and may utilize these functions as one of their bioenergetic pathways [79].

The metabolic functions related to the cobalt–zinc–cadmium efflux, nickel transport protein, iron transport protein, sulfite reductase, and the sulfur-oxidizing systems were abundant in the HGC samples. Our correlation analysis revealed that the cobalt/nickel efflux metabolism showed a significant positive correlation with the genera *Thiomonas*, *Metallibacterium*, and *Acidiphilium*, which were abundant in the HGC. The lower reaches of the HGC were observed to be contaminated with a high concentration of iron, sulfur, cobalt, nickel, and zinc because of the confluence of the geothermal valley [20]. The higher concentration of heavy metals, such as nickel and cobalt, causes the replacement of other essential metals and the perturbation of protein functions that are required for bacterial growth, leading to oxidative stress [80,81]. The efflux pump mechanism is one of the key mechanisms involved in regulating the internal metal ion concentrations of microbes [82]. Excess heavy metals are pumped out of the cytoplasm of microbes by metal transport and are further counterbalanced by the efflux system [83,84]. These pathways impart resistance to metal toxicity and provide a survival advantage to the bacterial communities living in heavily metal-contaminated water bodies, such as the HGC. Similarly, acidophilic sulfur-oxidizing bacterial communities abundant in hot springs can oxidize various reduced inorganic sulfur compounds to obtain electrons for their growth and survival [85]. The higher expression of the sulfur metabolism and arsenic detoxification could help bacterial communities adapt better to the sulfur-and arsenic-contaminated HGC samples. The higher expression of sulfur oxide-related pathways in the HGC is responsible for the oxidation of sulfide and thiosulfate to sulfate, which is used as an energy source [86]. Likewise, *Acidiphilium*, which is dominant in the HGC, exhibits the bifunctional enzyme CysN/CysC, which is one of the components of the assimilatory sulfate reduction pathways involved in reducing sulfate to sulfide [87]. The bifunctional enzyme, sulfane dehydrogenase subunit SO_X_C, which is significantly enriched in the HGC samples, is often associated with thermophilic bacterial communities that catalyze the reduction of elemental sulfur to hydrogen sulfide [88]. 

The PHC samples were enriched with the cobalt–zinc–cadmium efflux, thiosulfate/3-mercaptopyruvate sulfurtransferase, and high-affinity iron transportation system-related pathways. Our correlation results revealed a positive correlation between the cobalt efflux and the genus *Thiomonas*, which is found to be abundant in the PHC. The downstream area of the PHC may be contaminated with metals, such as zinc and cobalt, in the Bayan zone [15]. Therefore, a higher selection pressure for the expression of metal efflux functions was observed in the PHC, which is a bacterial survival strategy. The Bayan zone of the PHC has been shown to have a higher concentration of hydrogen sulfide [51], which might be involved in the sulfide oxidation pathway to generate thiosulfate [89]. Thiosulfate is an important intermediate that plays a key role in the bacteria-mediated sulfur-reducing processes [90]. The abundance of thiosulfate sulfurtransferase in the PHC samples catalyzes the metabolism of thiosulfate and oxidizes thiosulfate into other forms of sulfur substrates, which are utilized as an energy source for thriving bacterial communities [85]. Along with the sulfur metabolism, thiosulfate sulfurtransferase is also involved in the iron–sulfur cluster-forming metabolic pathway [91]. *Sulfuriferula* had a significant positive correlation with thiosulfate sulphurtransferase. It was abundant in the PHC samples, resulting in a higher expression of the iron–sulfur cluster, forming metabolic pathways in the PHC compared to the other watersheds. In addition, most of the uncultured bacterial genera were significantly and positively correlated with the predominant metabolic functions, including sulfur, iron, and the heavy metal efflux system-related pathways, indicating the existence of potentially novel bacteria in the three watersheds. However, further studies are warranted to identify these uncultured bacteria, and their roles, by simultaneously using third-generation sequencing and improved culture approaches. Additionally, the verification of the predicted bacterial functions using metabolomics and transcriptomics will provide better resolution than 16S rRNA amplicon sequencing.

## 5. Conclusions

Although numerous studies have been conducted on the hydrogeochemical characteristics of the TVG hot springs, our study is the first attempt to elucidate the influence of hot springs on the bacterial community structure of the proximal water bodies. This study provides important insights into bacterial abundance and the functional potential of the three water bodies of the TVG basin using next-generation sequencing. A higher bacterial diversity and abundance was observed in the NHC, followed by the HGC and the PHC. Despite the distinct bacterial abundance in each watershed, hot spring-related bacterial genera, such as *Sediminbacterium* and *Novosphingobium*, were common in the three sites. In addition, the abundance of *Bdellovibrio*, *Diplorickettsia*, and *Legionella* in the water samples indicated the influence of anthropogenic activities on the water bodies, which may pose a public health threat. The iron-and sulfur-related metabolic pathways were highly abundant in the study sites. However, heavy metal-related functions were dominant in the HGC and PHC, revealing that heavy metal contamination might shift the respective metabolic pathways. The bacterial community structure and the predicted functions demonstrated the influence of the hydrochemistry of the TVG hot springs on the associated water bodies. The results of this study may help to emphasize hot spring drainage management and help to frame the precise conservation and monitoring measures to protect proximal lotic water bodies.

## Figures and Tables

**Figure 1 microorganisms-10-00500-f001:**
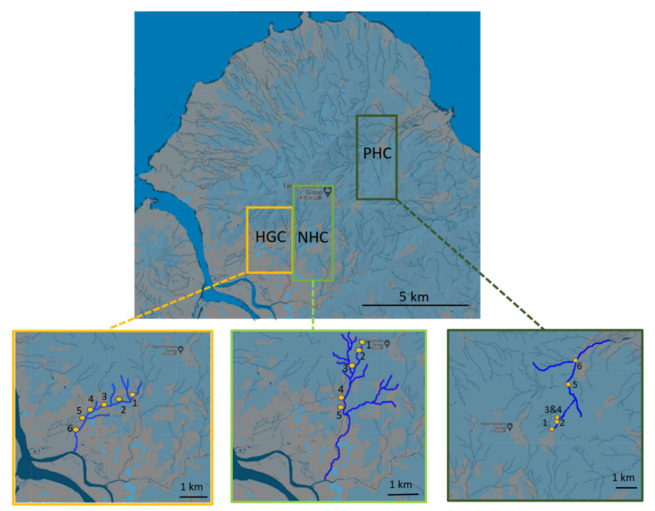
Distribution of sampling locations of HGC, NHC and PHC, TVG basin. The exact sampling site details are given in Appendix A.

**Figure 2 microorganisms-10-00500-f002:**
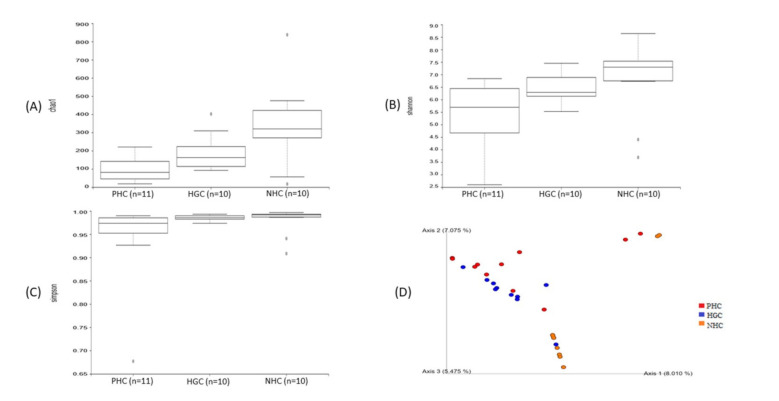
Box plot representing the bacterial alpha diversity and richness indices (**A**) Chao1, (**B**) Shannon index, and (**C**) Simpson index of the samples from the three watersheds. (**D**) PCA representing the beta diversity among the samples from the three watersheds.

**Figure 3 microorganisms-10-00500-f003:**
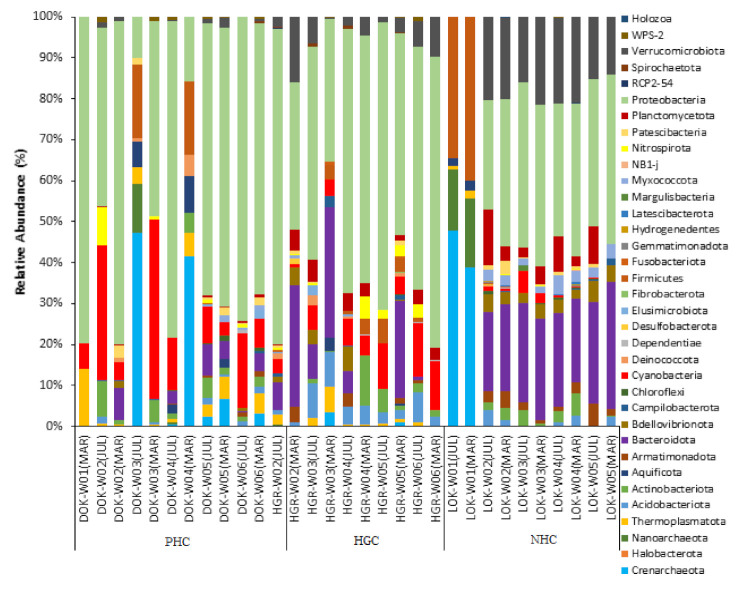
The stacked bar graphs illustrate the relative bacterial phyla abundance of the samples of three watersheds in two sampling months.

**Figure 4 microorganisms-10-00500-f004:**
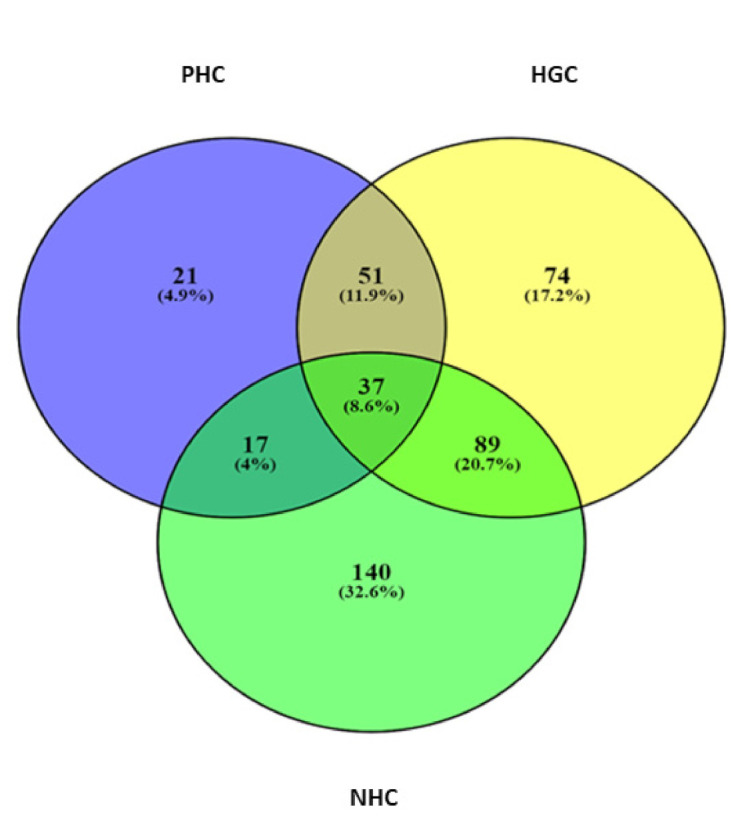
Venn diagram illustrating the proportions of unique and common bacterial genera noticed among the three watersheds.

**Figure 5 microorganisms-10-00500-f005:**
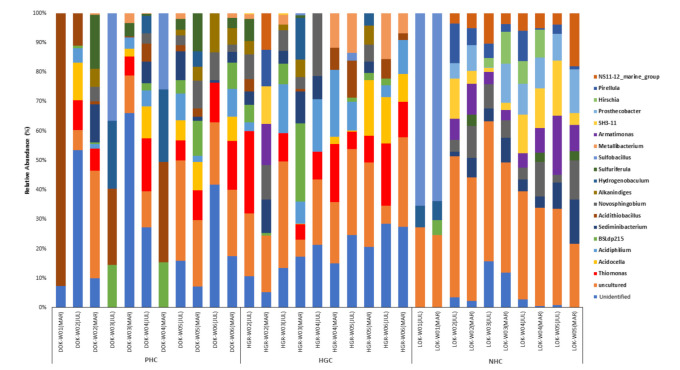
The stacked bar graphs illustrate the relative bacterial genera abundance of the samples of three watersheds in two sampling months.

**Figure 6 microorganisms-10-00500-f006:**
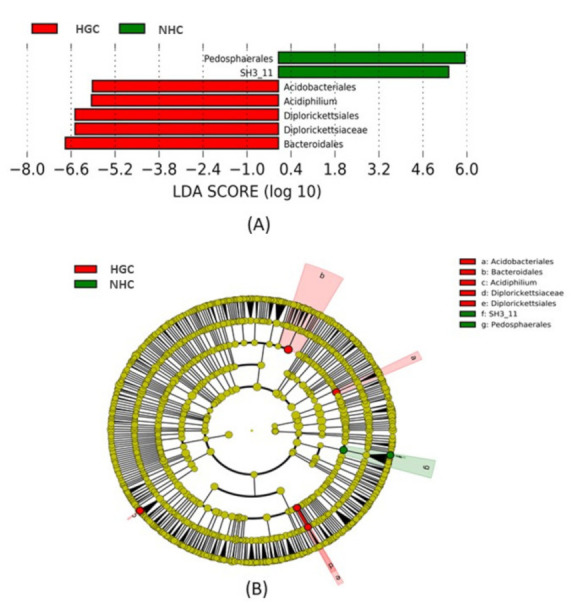
LDA combined with LEfSe showed the abundant bacterial clades. (**A**) Histogram of LDA score between the HGC and NHC samples. (**B**) LEfSe analysis of significant bacterial clades between the HGC and NHC samples.

**Figure 7 microorganisms-10-00500-f007:**
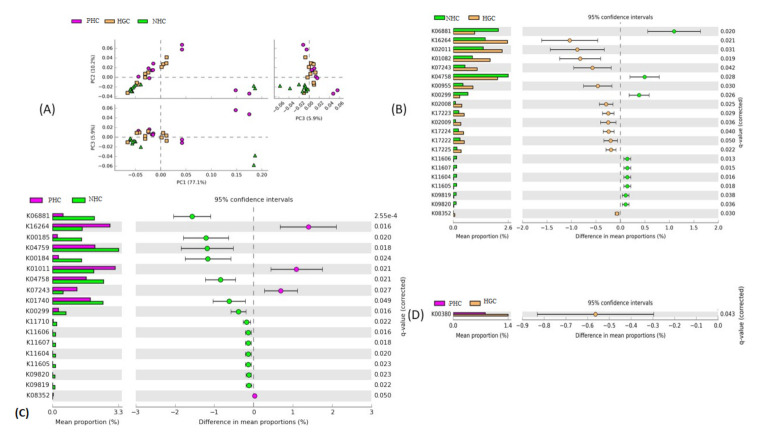
Functional prediction of the bacterial communities. (**A**) PCA showing the grouping of samples based on their predicted metabolic functions; significantly enriched bacterial functions among the (**B**) NHC and HGC, (**C**) PHC and NHC, and (**D**) PHC and HGC.

**Figure 8 microorganisms-10-00500-f008:**
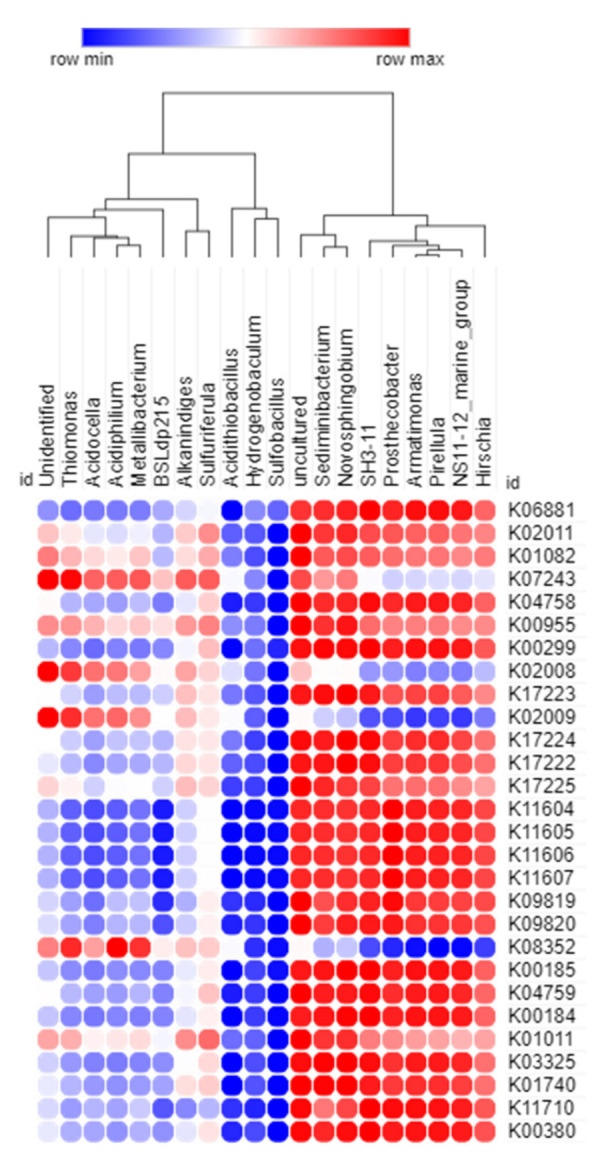
Spearman correlation analysis between bacterial genera of the three watersheds and their metabolic functional predictions. The positive and negative correlations are shown in red and blue colors, respectively.

## Data Availability

In a rigorously anonymous form, raw sequencing data have been deposited in the NCBI depository under the following ID: SUB10987666.

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
