# Peer review of "The Evaluation of Bacterial Abundance and Functional Potentials in the Three Major Watersheds, Located in the Hot Spring Zone of the Tatun Volcano Group Basin, Taiwan"

_microorganisms, 2022, doi:10.3390/microorganisms10030500_

Round 1

Reviewer 1 Report

The article is within the scope of the journal and deals with the taxonomic and functional microbial community in the hot spring zone of the Tatun Volcano Group basin, Taiwan. The samples were collected in spring and summer at several points along the thermal creek flow, which increased the reliability of the data. Statistical and correlation analyses revealed the differences in the phylogenetic diversity and metabolic potential between communities of different hydrothermal zones. Detection of pathogenic microorganisms indicated anthropogenic contamination of one of the basins. In general, the article may be published, although in our opinion some revision is required.

  1. While the authors provide references to the previous physicochemical and geochemical studies of the thermal springs and water catchment areas, the text contains no data on the actual physicochemical parameters of the samples (temperature, pH, Eh, concentrations of oxygen, sulfate, sulfide, FeII, FeIII, Mn, Ni, etc.). Without this information, discussion of the functional potential of the components of microbial communities do not seem well-founded. These data were probably present in the supplementary materials, which, however, we could not access.
  2. If the physicochemical parameters of the springs are indeed available, analysis of the correlation between the major parameters and the relative abundance of detected microbial groups would have been highly advantageous.
  3. The quality of Figures 2, 4, 5, and 6 is low, and it is impossible to decipher the major inscriptions.
  4. Careful editing is certainly recommended. An error occurring at the very beginning of the article (Abstract, line 24, where the word "genera" is certainly required, rather than "genus") makes the remaining text less trustworthy. Unfortunately, this is no the only one.

Author Response

We are grateful to the reviewer for his insightful comments on our paper. We have been able to include changes to reflect most of the suggestions provided by the reviewer.

Reviewer 2 Report

This is an interesting and well conducted work characterizing the bacterial communities of three major watersheds in Taiwan. The work follows all the standard methodology involved in the 16S rRNA study of bacterial communities and this was accompanied by a proper analysis of the local environmental data which allows to infer about the reasons explaining the community diversity. The objectives of the study are clear, and the results seem to highlight the major findings of this work. The discussion is sufficiently based on the obtained data, and the raised hypothesis seem reasonable. The number and choice of references are generally speaking adequate, however some references should be added in some sections (mentioned in the list of comments). Although the order of the sections of the manuscript are logical, and well introduced, the employed English language is sometimes not the most adequate.  

There are three major weaknesses of this manuscript which should be addressed beforehand in a major review: 

Firstly, all the figures were uploaded with low quality. Some figures such as figure 4 make it impossible to read and evaluate the corresponding data and therefore to evaluate properly this manuscript. Thus, the authors should upload new figures with higher resolution. 

Secondly, the authors should deposit the gene sequencing data of this study in public database. 

Lastly, the authors should also review the manuscript regarding the English language, since there are a significant number of mistakes, some of which were described in the attached list of comments, which also includes other minor comments. 

Author Response

(The authors gave the same response as above.)

Reviewer 3 Report

Dear Authors,

I have some suggestions and comments for your manuscript, following:

  • Line 25, 26, 242, 245  and others- Let  improve please, Chloroplast is not a phylum or genus of bacteria, but the site where plant photosynthesis takes place.
  • Line from 213-214: The sample W02 (not WO3) of PHC was found dominant with Nitrospirota and Actinobacteria, and W04 abundant in Crenarchaeota, Aquificota and Firmicutes.
  • 4, 5 and 6 is unreadable, please correct the font size.

Reviewer

Author Response

(The authors gave the same response as above.)

Round 2

Reviewer 1 Report

The authors made the necessary corrections and improved the quality of the illustrations. This version of the article can be published.

Author Response

Reviewer comments response

Dear editor,

We are grateful to the reviewer for his insightful comments on our paper. We have been able to include changes to reflect most of the suggestions provided by the reviewers.

Reviewer #1 Evaluations

Comments and Suggestions:  The authors made the necessary corrections and improved the quality of the illustrations. This version of the article can be published.

Response: Thank you for the comments and suggestion.

Reviewer 2 Report

I appreciate the authors letter and also the careful and meticulous answer to all the questions and comments that were raised. As a whole, the authors have done a good job addressing the major weaknesses of the previous version of the manuscript and the minor comments were also addressed in a satisfactory way. 

I only have a few minor comments regarding this new version of the manuscript. These are the following: 

Line 55: “because of anthropogenic influences.” Please replace with “due to anthropogenic”.

Line 155: “400 nM”.  This font size is too big here.

Line 244: “common bacterial genus genera”. Please rephrase.

Line 263: “abundance between the samples”. It should be abundance of the samples

Line 752-753: This reference has no journal. 

Line 769-70: There is an error in the formatting style (extra space for "259-287").
